# Longitudinal Associations Between Maternal Anemia and Breastfeeding Practices on Infant Hemoglobin Levels in the Lao People’s Democratic Republic

**DOI:** 10.3390/nu17101703

**Published:** 2025-05-16

**Authors:** Stephanie M. Khoury, Najmeh Karimian-Marnani, Souliviengkham Sonephet, Günther Fink, Jordyn T. Wallenborn

**Affiliations:** 1University of Basel, 4001 Basel, Switzerland; najmeh.karimian-marnani@swisstph.ch (N.K.-M.); souliviengkham.sonephet@swisstph.ch (S.S.); guenther.fink@swisstph.ch (G.F.); jordyn.wallenborn@swisstph.ch (J.T.W.); 2Department of Epidemiology and Public Health, Swiss Tropical and Public Health Institute, 4123 Allschwil, Switzerland; 3Lao Tropical and Public Health Institute, Vientiane, Laos

**Keywords:** anemia, hemoglobin, breastfeeding, pregnancy, postpartum, maternal, child, health

## Abstract

Background: Anemia is a chronic condition that disproportionately affects women and children. Anemia shows detrimental long-term impacts on maternal and child health and development, with the highest burden observed in low- and middle-income countries. In the Lao People’s Democratic Republic (PDR), anemia is prevalent in 39% of women of reproductive age, 47% of pregnant women, and 43% of children under five. Objective: Our study evaluates associations between maternal anemia at pregnancy and postpartum and infant hemoglobin (Hb) levels in early infancy. We further investigate the influence of breastfeeding practices on this association. Methods: Data from the Social Transfers for Exclusive Breastfeeding ongoing randomized control trial (RCT) (*n* = 298) in Vientiane, Lao PDR, was used. Maternal and infant Hb levels were assessed during pregnancy and at one, six, and twelve months postpartum. We used generalized estimating equations (GEE) for repeated measures analysis. Results: Anemic mothers at pregnancy and postpartum saw a 1.89 g/L (adjusted 95% CI: −4.48 to 0.70, *p* = 0.15) and 3.25 g/L (adjusted 95% CI: −7.86 to 1.36, *p* = 0.17) decrease in infant Hb levels compared to non-anemic mothers. Among postpartum anemic mothers who followed the World Health Organization (WHO) breastfeeding recommendations, an increase of 7.87 g/L in infant Hb levels (adjusted 95% CI: −2.21 to 17.94, *p =* 0.13) was observed. Conclusions: A weak negative association was found between maternal anemia during pregnancy and the first year postpartum and infant Hb levels. However, among anemic mothers at postpartum, adhering to WHO breastfeeding recommendations may help to mitigate this impact.

## 1. Introduction

Anemia, a chronic health condition characterized by insufficient hemoglobin (Hb) levels, remains a significant global public health challenge. Anemia mainly affects women of reproductive age, pregnant women, and young children, with the highest prevalence observed in low- and middle-income countries (LMICs) [1]. The most common symptoms include chronic fatigue, dizziness, and shortness of breath; if left untreated, anemia can result in more severe and long-term debilitating effects [2]. In pregnant women, anemia is associated with an increased risk of maternal morbidity, preterm birth, and low birth weight, while infants born to anemic mothers are at higher risk of physical and cognitive developmental delays, impaired immunity, and behavioral changes [2,3]. The World Health Organization (WHO) estimates that approximately 37% of pregnant women and 40% of children under five globally are affected by anemia [4].

Anemia is a serious health concern in the Southeast Asian region, where cultural practices, dietary habits, and limited healthcare access contribute to its prevalence and challenges in its management [5]. In the Lao People’s Democratic Republic (PDR), the prevalence of anemia is alarmingly high, with 47% of pregnant women and 43% of children under five being anemic [6,7,8]. Traditional remedies are often preferred over medically prescribed iron supplements, which can delay effective treatment [9]. Economic constraints also limit the consumption of iron-rich foods, such as meat and green leafy vegetables [10]. Additionally, rural areas have limited access to healthcare facilities, and most of these facilities do not have Hb tests available, affecting the timely diagnosis and treatment of anemia [11].

Pregnant women face unique risks of developing anemia due to physiological changes that increase plasma volume relative to red blood cell mass [12]. Anemia during pregnancy not only affects the mother’s health but also has significant implications for fetal development and infant health outcomes [13]. Due to insufficient micronutrient transfer from mother to fetus during pregnancy, infants are at higher risk of developing micronutrient deficiencies, particularly iron deficiency, which is closely associated with anemia [14]. Maternal anemia during pregnancy may also lead to postpartum anemia, which can potentially impact infant health outcomes [15]. To date, few studies have followed mother–infant dyads from pregnancy through the postpartum period to explore how maternal anemia affects infants’ Hb levels in their first years of life.

Breastfeeding is an essential component of infant nutrition and health. Breast milk provides infants with key nutrients, including iron, which are vital for infant growth and development. According to the WHO, infants should be exclusively breastfed for the first six months, and complementary breastfed until two years of age [16]. While breastfeeding has well-documented benefits for both maternal and child health, anemic mothers may face challenges such as reduced milk supply, which can indirectly affect the infant’s iron intake [17]. However, little research has been carried out on how maternal anemia and breastfeeding practices interact to influence infant iron levels, particularly in the context of the Lao PDR.

This study aims to explore the relationship between maternal anemia and infant Hb levels in the Lao PDR. By following mother–infant dyads from pregnancy through the first year postpartum, this research study seeks to provide insights into the relationship between maternal anemia and infant Hb levels, while also assessing how breastfeeding practices can alter this relationship. Our findings aim to inform public health policies and interventions that address the dual burden of anemia in mothers and infants in the Lao PDR.

## 2. Materials and Methods

### 2.1. Data Source

We used data from the ongoing Vientiane Multi-Generational Birth Cohort (VITERBI) and the Social Transfer for Exclusive Breastfeeding (STEB) randomized control trial (RCT) [18,19]. The VITERBI cohort includes 3000 mothers from four districts in the capital city of the Lao PDR, Vientiane: Chanthabuly, Sikhottabong, Sangthong, and Pakngum. Each district was strategically chosen to represent the two highest and the two lowest socioeconomic status districts and to equally distribute participants between urban and rural settings. STEB is an RCT and prospective cohort nested within VITERBI running from August 2022 to August 2026, which assesses the effect of a social transfer intervention on EBF rates and the long-term impacts on both maternal and child health in the first three years postpartum.

Mothers were selected for STEB from the VITERBI cohort based on the following inclusion criteria: had a due date or gave birth between 1 July 2022 and 30 June 2023; gave birth within the last four weeks prior to inclusion start; were exclusively breastfeeding at the time of recruitment; had no illnesses that contraindicated breastfeeding; and had a healthy singleton infant of 37 weeks or more gestation with a birth weight of at least 2500 g.

STEB participants (*n* = 298) were randomly assigned to one of three groups: a control group that received breastfeeding education only; an intervention group receiving breastfeeding education and an unconditional social transfer at six months postpartum; and an intervention group receiving breastfeeding education with a social transfer, conditional upon still exclusive breastfeeding at six months postpartum. The social transfer included either (1) cash, (2) diapers, (3) baby clothes, or (4) development toys, as selected by the participants themselves. Each option could be chosen individually or as a combination, as long as it did not exceed the pre-determined value of USD 75 per participant.

Participants were asked to provide oral consent and sign an informed consent form prior to being included in the study. They were interviewed at one month postpartum (between July 2022 and April 2023), six months postpartum (between December 2022 and October 2023), and twelve months postpartum (between June 2023 and March 2024). Demographics, labor and delivery, maternal and infant health measurements, and infant feeding information were collected from questionnaires, maternal and infant measurements, and biospecimen samples. Figure 1 illustrates data collection and sample sizes across all study time points. Eighteen cases were lost to follow-up at six months postpartum (*n* = 280), and thirty-four were not assessed at twelve months postpartum (*n* = 266). Additionally, twenty-nine cases had missing information (*n* = 237), including participants lost to follow-up at six months who returned later at twelve months postpartum (*n* = 14), participants with missing Hb levels at any time point (*n* = 12), and infants with missing Hb measurements (*n* = 3).

To avoid inaccurate self-reports of breastfeeding, stemming from social desirability bias in participants, the research team employed additional techniques to reduce these biases: speaking with family members about supplementation, looking around the house for formula, and having participants provide human milk samples.

### 2.2. Infant Hemoglobin: Outcome Variable

Given that our study assessed the longitudinal impact of maternal anemia on the infant, our main outcome of interest was infant Hb levels, measured at one month, six months, and twelve months postpartum. The infant Hb levels in g/L were taken using the HemoCue Hb 301, which determines the Hb concentration by measuring the absorbance of whole blood at a Hb/HbO_2_ isosbestic point [20]. We considered categorizing infants into whether they were anemic or not, based on literature-specified cutoffs [21,22]; however, anemia cutoffs for infants less than six months are not defined. Therefore, infant Hb was taken as a continuous measurement as the primary outcome variable.

### 2.3. Maternal Anemia: Exposure Variable

The main exposure of interest was maternal anemia (anemic or not anemic) measured at four time points: pregnancy, one month postpartum, six months postpartum, and twelve months postpartum. Maternal Hb levels were taken using the HemoCue Hb 301 instrument [20]. We use WHO Hb level cutoffs to define maternal anemia [21]. During pregnancy, participants with Hb levels of less than 110 g/L were classified as anemic, whereas those with greater than or equal to 110 g/L were classified as not anemic [23]. During the postpartum period, maternal Hb levels less than 120 g/L were categorized as anemic, whereas participants with Hb levels greater than or equal to 120 g/L were classified as not anemic [21]. Hb levels were used to assess anemia status for both mother and infant as other biomarkers were unavailable. Although not always a reliable indicator, the WHO still recommends their use for assessing anemia, provided that standardized guidelines are followed [24].

### 2.4. Compliance with Breastfeeding Recommendations

All participants were classified into three groups based on their compliance with WHO breastfeeding recommendations [16,25]. Participants were categorized as “followed recommendations” if they exclusively breastfeed for six months postpartum and reported to be still breastfeeding at twelve months postpartum. Despite the WHO recommending breastfeeding up to two years [16], we categorized “followed recommendations” as those that breastfeed for twelve months, given that two-year breastfeeding data were not yet collected. Participants that “partially followed recommendations” were exclusively breastfeeding for at least four months postpartum and still breastfeeding at twelve months postpartum. Participants were categorized as “did not follow recommendations” if they were exclusively breastfeeding for less than four months, or were not breastfeeding until twelve months postpartum.

It is important to note that all women were exclusively breastfeeding for the first month postpartum. The categorization of the participants within different breastfeeding recommendation groups is depicted in a supplementary figure found in Appendix A.

### 2.5. Statistical Analysis

Descriptive statistics were used to obtain frequencies and percentages for categorical variables, and mean and standard deviations (SDs) for continuous variables. Pearson’s Chi-squared test was used to identify significant differences between the groups that followed, partially followed, or did not follow the breastfeeding recommendations. To help visualize the data, we used a bar graph to illustrate the impact of maternal anemia on infant Hb levels at the different time points.

To accommodate the longitudinal analysis in our study, we used generalized estimating equations (GEEs) models to estimate the associations between repeated measures (one, six, and twelve months) of maternal anemia at postpartum (time-varying binary variable) and infant Hb levels (time-varying continuous variable) observed at the same time points. Additionally, we included maternal anemia at pregnancy (binary) as a predictor variable to the models. We included an interaction between postpartum anemia and time to investigate effect modification over time. We initially further investigated effect modification for iron supplementation at pregnancy and breastfeeding practices; however, due to small sample sizes of women who did not have iron supplementation, we did not include this as an interaction term.

For the GEE models, we used unstructured correlation structure given that the correlations between the measurements varied freely and the “identity” link, since the responses for infant Hb were normally distributed [26,27]. We stratified GEE models by the participant’s classification of whether they followed the WHO breastfeeding recommendation. Given the small sample size available, we opted against specific *p*-value cutoffs for significance, and simply show 95% CIs and *p*-values for all estimates.

We adjusted for other confounders identified in the literature that are associated with the outcome [28,29], and which changed the GEE coefficient estimate of anemia by >10%. Confounders included maternal age, sex of the baby, district, education status, marital status, household quintile, iron supplements at pregnancy, maternal body mass index (BMI) at pregnancy and maternal alcohol consumption during pregnancy.

Data analysis was performed using R statistical computing environment, using “tidyverse” for data manipulation, and “geepack” for GEE model construction [30,31].

## 3. Results

### 3.1. Demographics

Demographic characteristics of participants who completed their one, six- and twelve month follow-up visits are displayed in Table 1 (*n* = 252). The mean age of participants was 27.4, the majority were married or cohabitating (92%), and had attained at least secondary education (40%) or tertiary education (34%). The district of Sikhottabong was the most represented in the cohort (42%), followed by Sangthong (24%) and Pakngum (24%). Nearly all women took iron supplements during pregnancy (91%) and did not consume alcohol during pregnancy (77%).

### 3.2. Infant Hemoglobin Trend over Time

Figure 2 depicts the mean infant Hb levels at three postpartum time points, stratified by maternal anemia status: ever had anemia (at pregnancy or any point postpartum) versus those who never had anemia. The graph shows higher infant Hb levels among mothers who were never anemic; however, this difference diminishes overtime.

### 3.3. Association Between Maternal Anemia and Infant Hb Levels

Table 2 and Table 3 display the unadjusted and fully adjusted associations, respectively, between maternal postpartum anemia and infant Hb levels at one, six, and twelve months postpartum; all models were stratified by whether the participant followed the WHO breastfeeding recommendations.

The overall adjusted GEE model revealed that if the mother was anemic at pregnancy, infant Hb levels decreased by 1.89 g/L over time, compared to infants of non-anemic mothers (adjusted β = −1.89, 95% CI −4.48, 0.70, *p =* 0.15; Table 3). If the mother was anemic at a given postpartum time point, infant Hb levels decrease by 3.25 g/L at the same time point, compared to infants of non-anemic mothers (adjusted β = −3.25, 95% CI −7.86, 1.36, *p =* 0.17; Table 3).

Among mothers who followed the WHO breastfeeding recommendation, infants’ Hb levels were 4.56 g/L lower if the mother was anemic during pregnancy, compared to infants of non-anemic mothers (adjusted β = −4.56, 95% CI −10.73, 1.62, *p =* 0.15; Table 3). Surprisingly, among mothers who were anemic at postpartum and followed the recommendation, the infant’s Hb levels were 7.87 g/L (adjusted β = 7.87, 95% CI −2.21, 17.94, *p =* 0.13; Table 3) higher compared to infants of non-anemic mothers.

Among mothers who only partially followed the WHO breastfeeding recommendations, being anemic during pregnancy lowered infant Hb levels by 2.52 g/L compared to mothers who were not anemic (adjusted β = −2.52, 95% CI −7.27, 2.24, *p =* 0.30; Table 3). Infants of anemic mothers at postpartum who partially followed the recommendation had, on average, 13.30 g/L lower Hb levels (adjusted β = −13.30, 95% CI −22.61, −3.99, *p =* 0.01; Table 3), compared to infants of non-anemic mothers.

Among mothers who did not follow the WHO breastfeeding recommendations, being anemic during pregnancy led to a 0.61 decrease in infant Hb levels compared to mothers who were not anemic (adjusted β = −0.61, 95% CI −4.12, 2.90, *p =* 0.73; Table 3). Infants of anemic mothers at postpartum who did not follow the recommendation showed a 3.58 g/L lower infant Hb levels compared to infants of mothers who were not anemic (adjusted β = −3.58, 95% CI −9.35, 2.19, *p =* 0.22; Table 3).

We observe a decreasing trend in infant Hb levels from one to six months (adjusted β = −14.94, 95% CI −18.47, −11.42, *p* < 0.001; Table 3), and from one to twelve months (adjusted β = −16.64, 95% CI −20.53, −12.75, *p* < 0.001; Table 3). There is no evidence of effect modification by maternal postpartum anemia over time (Table 3).

## 4. Discussion

Our study assessed the longitudinal relationship between maternal anemia during pregnancy through the first year postpartum and infant Hb levels, at one, six, and twelve months postpartum. We further evaluated whether following the WHO breastfeeding recommendations of exclusive breastfeeding for six months after birth and continuing complementary breastfeeding thereafter altered this relationship. Our findings suggest that maternal anemia during pregnancy and the postpartum period is associated with lower infant Hb levels. The anemic status of mothers during pregnancy who followed the breastfeeding recommendations seemed to have no effect on infant Hb levels. However, the infants of postpartum anemic mothers who followed the breastfeeding recommendations showed an increase in Hb levels. It is important to note that despite the wide CIs observed in our results, these explanatory analyses still provide valuable insights about maternal anemia and its effect on Hb levels at infancy.

To our knowledge, our study is the first to investigate the associations between maternal anemia and infant Hb levels longitudinally during pregnancy as well as the first year postpartum. Although some studies have assessed the impact of maternal anemia on infant Hb levels over time, maternal anemia was only measured either during pregnancy or postpartum. Studies assessing the relationship between anemia during pregnancy and infant Hb levels indicate that infants born to anemic mothers are more susceptible to developing anemia in the first years of life compared to those born to non-anemic mothers, in agreement with our findings [32,33]. A study conducted in China (*n* = 17,193) that assessed maternal anemia at 24–28 weeks of gestation found a negative impact on infant Hb levels at 5–7 and 11–13 months of age [32]. Similarly, another cohort study in India (*n* = 941) found that maternal anemia negatively affects infant Hb levels, though that was assessed in infants at two years of age [33]. More specifically, Abioye et al. found that the infant’s anemia risk was only affected if the mother specifically had iron deficiency anemia [34].

Our findings were consistent with other studies assessing postpartum maternal anemia and its impact on infant Hb levels [35,36]. A study of 183 Mexican infants revealed that the development of postpartum anemia at birth, three, and six months was associated with an elevated risk of anemia in infants at nine months [35]. Another study assessed maternal anemia at one month postpartum and found a decrease in infant Hb levels at birth and at fourteen weeks of age compared to those with non-anemic mothers [36].

In addition to assessing the longitudinal relationship between maternal anemia and infant Hb levels, we explored how breastfeeding practices influenced this relationship. Research on breastfeeding practices and the risk of infant anemia has been inconsistent. Some studies suggest that infants with anemic mothers who are exclusively breastfed for the first six months are not at an increased risk of iron deficiency compared to infants of non-anemic mothers, and may be protected against developing anemia, consistent with our findings [37,38,39]. Contrary to our findings and initial hypotheses, studies have found that anemic mothers who exclusively breastfeed for more than four months can negatively affect their infant Hb levels due to insufficient iron availability in breast milk [35,40,41]. A study in Nigeria further showed that Hb levels are inversely correlated with age when breastfeeding is stopped. Therefore, continued breastfeeding or adherence to strict breastfeeding guidelines may not be the best approach in determining anemia [42].

The low Hb levels found in infants born to anemic mothers during pregnancy can be attributed to the inadequate iron transfer from the mother to the fetus in utero [43,44]. This deficiency may result in lower iron stores in infants at birth, affecting their Hb levels during their first year of life. Additionally, anemia during pregnancy can lead to depleted iron stores during the lactation period, reducing milk production [36]. Although breastmilk typically provides sufficient nutrients, inadequate breastfeeding due to low milk supply may limit infants’ exposure to sufficient iron, which can explain why infant Hb levels are lower among anemic mothers [17,45].

The relationship between postpartum maternal anemia and infant Hb levels is more complex due to other contributing factors, such as the natural decline in infant Hb levels over time, whether the mother was anemic or not, and the differences in feeding practices [17,46]. When assessing the role of the WHO breastfeeding recommendations, we found that infants with anemic mothers at postpartum who follow the breastfeeding recommendations had higher Hb levels. This indicates that although iron levels in breastmilk naturally decline from birth to six months, breastmilk remains sufficient to support infant iron stores and provides essential nutrients that protect infant health and development during this period [37,45]. It is unclear why the group that partially followed the recommendations demonstrated a strong negative association with infant Hb levels, whereas the group that did not follow showed only a slight negative effect. We hypothesize that this discrepancy may stem from our small sample size and insufficient data on feeding behaviors and intensity, such as the frequency of breastfeeding, which would provide insight into the amount of breastmilk the infant received compared to other liquids and solids. Understanding how often the infant was breastfed per day and the ratio between breastfeeding and other food sources could influence iron intake and affect infant Hb levels. Additionally, knowing what supplementary milks, liquids, and solids were introduced during complementary feeding could offer additional insight into the infant’s diet and help account for dietary differences, including variations in iron intake from other nutritional sources.

Despite our *p*-values being above conventional significance thresholds, the study emphasizes the importance of diagnosing and treating anemia during pregnancy and postpartum to reduce the risk of anemia in infants. Healthcare professionals should routinely check maternal Hb levels and, if levels fall below the recommended Hb cutoff, determine the underlying causes and treat her accordingly [21]. Potential contributing factors to maternal anemia include the presence of infectious diseases, neglected tropical diseases, or micronutrient deficiencies [47]. Additionally, healthcare providers should monitor infant feeding practices, as these may also have a risk of infant anemia. Effective and adequate breastfeeding should be assessed, and once complementary foods are introduced, the quality and iron content of the child’s diet should be carefully observed [48,49].

Our study had some limitations. Our study population was restricted to healthy, singleton, term infants from Vientiane. Consequently, the findings may not be generalizable to the broader population of the Lao PDR. Having a relatively small sample size may have also reduced the validity of our results, particularly when interpreting the stratification by adherence to the WHO breastfeeding recommendations. The data on maternal and infant anemia could have been enhanced by including measurements of serum ferritin levels. Assessing serum ferritin would have allowed for a more accurate evaluation of iron stores in the body, and when combined with Hb levels, it could have provided a more comprehensive understanding of anemia status in both mothers and infants. However, serum ferritin levels were not measured in the primary study due to limited resources available within the study context. As already mentioned, we also omitted breastfeeding intensity questions, which could have provided additional insight into the observed discrepancies between the breastfeeding adherence groups. We also lacked a detailed food intake diary for both maternal and infant diets, which could have influenced our study outcome. Due to time constraints, the diet questions we included were limited to the consumption of fruits, vegetables, rice, potatoes, noodles, fermented foods, fried foods, red meat, and fish in the last seven days. Another limitation was the insufficient data on infant iron supplementation. While we asked the participants about infant iron supplementation, the majority did not respond or were unsure, reducing the reliability of this data in our analysis. Overall, the small sample size and missing data may affect the interpretation of our findings and should be addressed in future research.

## 5. Conclusions

In conclusion, this study is the first to examine the impact of longitudinal maternal anemia during pregnancy and postpartum on the Hb levels of infants up to twelve months of age, while highlighting the role of varying breastfeeding practices in shaping this relationship. We are also the first to assess this relationship in the Lao PDR and the Southeast Asian region. Our findings suggest that maternal anemia during pregnancy and postpartum increases the risk of infants developing anemia in their first year of life, but when postpartum anemic mothers follow WHO breastfeeding recommendations, it may help mitigate this risk. Health practitioners should monitor maternal Hb levels in regular health checkups to reduce the risk of infant anemia development. Further research is needed to better understand the relationship between maternal and infant anemia and in turn, reduce maternal anemia rates, and provide optimal conditions for infants to thrive during their first year of life. Furthermore, personalized postpartum management for mothers with anemia throughout the lactation period can help ensure sufficient breastfeeding while monitoring Hb levels in both mothers and infants.

## Figures and Tables

**Figure 1 nutrients-17-01703-f001:**
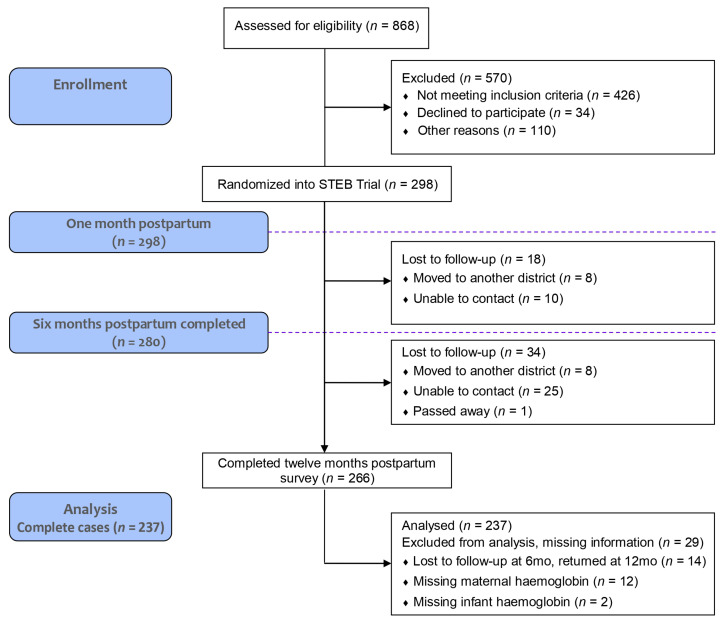
Study participant disposition. This figure describes the flow of participants throughout the VITERBI and STEB studies, as well as the complete cases used for the data analysis.

**Figure 2 nutrients-17-01703-f002:**
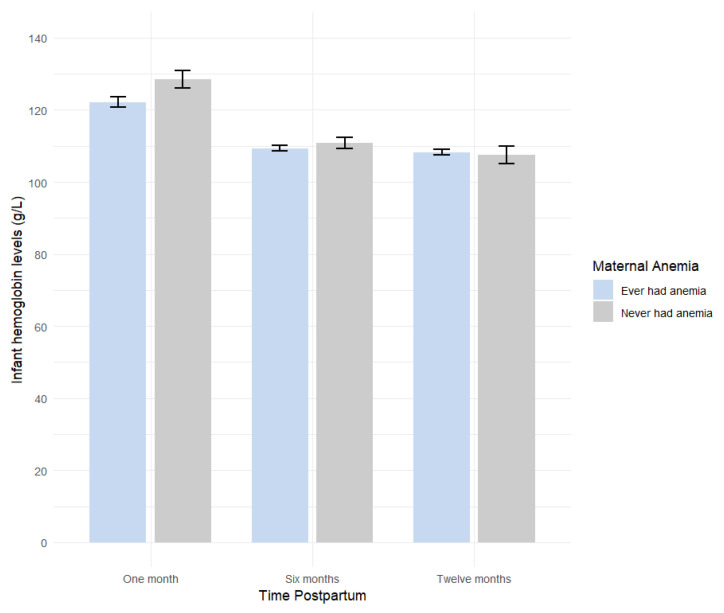
Mean infant hemoglobin (Hb) levels at one, six, and twelve months by maternal anemia status. Average infant Hb levels in g/L by time postpartum and whether the mother had anemia (ever anemic: mothers with anemia during pregnancy or postpartum; never anemic: mothers without anemia).

**Table 1 nutrients-17-01703-t001:** Pregnancy characteristics of study participants who completed the one, six, and twelve month postpartum visits.

Characteristic	Overall (*n* = 252)	WHO Breastfeeding Recommendations			*p*-Value ^1^
		Followed (*n* = 43)	Partially Followed (*n* = 59)	Did Not Follow (*n* = 150)	
	n (%) or mean (s.d.) ^2^				
Age (years) ^2^	27.4 (5.5)	27.6 (6.0)	27.3 (5.8)	27.4 (5.2)	0.21
Age (categorized)					0.95
Younger than 25	74 (29.7)	12 (28.6)	19 (33.3)	43 (28.6)	
Between 25 and 30	88 (35.3)	14 (33.3)	20 (35.1)	54 (36.0)	
Older than 30	87 (34.9)	16 (38.1)	18 (31.6)	53 (35.3)	
Marital Status					0.95
Married or cohabitating	228 (91.6)	39 (92.9)	52 (91.2)	137 (91.3)	
Not Married	21 (8.4)	3 (7.1)	5 (8.8)	13 (8.7)	
Maternal education					0.69
Primary or no education	64 (25.7)	14 (33.3)	13 (22.8)	37 (24.7)	
Secondary	100 (40.2)	15 (35.7)	26 (45.6)	59 (39.3)	
Tertiary	85 (34.1)	13 (31.0)	18 (31.6)	54 (36.0)	
District in Vientiane					0.41
Chanthabuly	26 (10.8)	1 (2.4)	7 (12.3)	18 (12.0)	
Pakngum	57 (23.7)	13 (30.9)	15 (26.3)	33 (22.0)	
Sangthong	58 (24.0)	12 (28.6)	16 (28.1)	34 (22.7)	
Sikhottabong	100 (41.5)	16 (38.1)	19 (33.3)	65 (43.3)	
Household wealth index					0.35
1st quartile	50 (19.8)	13 (30.2)	12 (20.3)	25 (16.7)	
2nd quartile	45 (17.9)	9 (20.9)	12 (20.3)	24 (16.0)	
3rd quartile	52 (20.6)	6 (14.0)	15 (25.4)	31 (20.7)	
4th quartile	72 (28.6)	10 (23.3)	16 (27.1)	46 (30.7)	
5th quartile	33 (13.1)	5 (11.6)	4 (6.78)	24 (16.0)	
Iron supplementation					0.63
Yes	226 (90.8)	39 (92.9)	53 (93.0)	134 (89.3)	
No	23 (9.2)	3 (7.14)	4 (7.0)	16 (10.7)	
Alcohol intake					0.92
Yes	57 (22.9)	10 (23.8)	14 (24.6)	33 (22.0)	
No	192 (77.1)	32 (76.2)	43 (75.4)	117 (78.0)	
Maternal BMI ^2^	25.0 (6.3)	24.8 (4.6)	24.4 (4.6)	25.3 (7.3)	0.50

World Health Organization (WHO) breastfeeding guidelines: “Followed” = mother exclusively breastfed infant for six months and infant still breastfeeding at twelve months postpartum; “Partially Followed” = exclusive breastfeeding for four months and still breastfeeding at twelve months postpartum; “Did Not Follow” = either not exclusively breastfeeding at four months postpartum, or exclusively breastfeeding for four months but no longer breastfeeding at twelve months postpartum. ^1^ Pearson’s Chi-squared test was used to obtain *p*-values for statistical differences between categories; ^2^ The variable is continuous and the values are shown as a mean, with the standard deviation in brackets.

**Table 2 nutrients-17-01703-t002:** Association between maternal anemia at pregnancy and postpartum and infant hemoglobin (Hb) levels at one, six, and twelve months using unadjusted generalized estimating equation models.

Variable	Overall		WHO Breastfeeding Recommendations					
			Followed		Partially Followed		Did Not Follow	
	β (95% CI)	*p*	β (95% CI)	*p*	β (95% CI)	*p*	β (95% CI)	*p*
Anemic at Pregnancy	−0.92 (3.30, 1.47)	0.45	−4.13 (−8.50, 0.24)	0.06	−0.30 (−4.76, 4.16)	0.89	0.09 (−3.25, 3.44)	0.96
Anemia at PP	−3.30 (−7.99, 1.39)	0.17	7.34 (−3.41, 18.08)	0.18	−13.09 (−22.47, −3.70)	0.01	−3.11 (−8.94, 2.72)	0.30
Time								
One month PP	Reference							
Six months PP	−14.97 (−18.52, −11.42)	0.00	−11.80 (−19.81, −3.80)	0.00	−15.99 (−22.84, −9.13)	0.00	−15.92 (−20.69, −11.15)	0.00
Twelve months PP	−16.63 (−20.53, −12.74)	0.00	−17.13 (−24.59, −9.68)	0.00	−21.86 (−31.42, −12.30)	0.00	−13.99 (−18.59, −9.39)	0.00
PP Anemia × Time Interaction								
Anemia at one month PP	Reference							
Anemia at six months PP	1.97 (−3.38, 7.32)	0.47	−6.06 (−18.88, 6.77)	0.35	13.52 (2.31, 24.72)	0.02	1.56 (−5.11, 8.22)	0.65
Anemia at twelve months PP	2.47 (−3.03, 7.98)	0.38	−6.36 (−18.04, 5.32)	0.29	14.94 (2.39, 27.49)	0.02	−0.05 (−6.49, 6.40)	0.99

The time variable as an independent predictor of our outcome was included in the table for interpretation of the interaction term (PP anemia × time); World Health Organization (WHO) breastfeeding guidelines: “Followed” = mother exclusively breastfed infant for six months and infant still breastfeeding at twelve months postpartum; “Partially Followed” = exclusive breastfeeding for four months and still breastfeeding at twelve months postpartum; “Did Not Follow” = either not exclusively breastfeeding at four months postpartum, or exclusively breastfeeding for four months but no longer breastfeeding at twelve months postpartum; robust standard errors (SEs) are depicted in brackets following the estimates; PP = postpartum; CI = confidence interval.

**Table 3 nutrients-17-01703-t003:** Association between maternal anemia at pregnancy and postpartum and infant hemoglobin (Hb) levels at one, six, and twelve months using generalized estimating equation models fully adjusted for confounders.

Variable	Overall		WHO Breastfeeding Recommendations					
			Followed		Partially Followed		Did Not Follow	
	β (95% CI)	*p*	β (95% CI)	*p*	β (95% CI)	*p*	β (95% CI)	*p*
Anemic at Pregnancy	−1.89 (−4.48, 0.70)	0.15	−4.56 (−10.73, 1.62)	0.15	−2.52 (−7.27, 2.24)	0.30	−0.61 (−4.12, 2.90)	0.73
Anemia at PP	−3.25 (−7.86, 1.36)	0.17	7.87 (−2.21, 17.94)	0.13	−13.30 (−22.61, −3.99)	0.01	−3.58 (−9.35, 2.19)	0.22
Time								
One month PP	Reference							
Six months PP	−14.94 (−18.47, −11.42)	0.00	−12.13 (−19.87, −4.40)	0.00	−15.90 (−22.86, −8.94)	0.00	−15.97 (−20.73, −11.21)	0.00
Twelve months PP	−16.64 (−20.53, −12.75)	0.00	−15.50 (−23.74, −9.27)	0.00	−22.45 (−32.09, −12.81)	0.00	−14.20 (−18.81, −9.59)	0.00
PP Anemia × Time Interaction								
Anemia at one month PP	Reference							
Anemia at six months PP	1.90 (−3.46, 7.26)	0.49	−4.94 (−17.72, 7.85)	0.45	13.26 (1.83, 24.69)	0.02	1.55 (−5.15, 8.26)	0.65
Anemia at twelve months PP	2.49 (−3.00, 7.98)	0.37	−7.71 (−18.99, 3.57)	0.18	16.32 (3.52, 29.13)	0.01	0.37 (−6.11, 6.85)	0.91

The time variable as an independent predictor of our outcome was included in the table for interpretation of the interaction term (PP anemia × time); World Health Organization (WHO) breastfeeding guidelines: “Followed” = mother exclusively breastfed infant for six months and infant still breastfeeding at twelve months postpartum; “Partially Followed” = exclusive breastfeeding for four months and still breastfeeding at twelve months postpartum; “Did Not Follow” = either not exclusively breastfeeding at four months postpartum, or exclusively breastfeeding for four months but no longer breastfeeding at twelve months postpartum; All GEE models were adjusted for maternal age, sex of the baby, district, maternal education status, marital status, household quintile, iron supplements at pregnancy, maternal BMI at pregnancy, and maternal alcohol consumption during pregnancy; robust standard errors (SEs) are depicted in brackets following the estimates; PP = postpartum; CI = confidence interval.

## Data Availability

Data described in the manuscript, codebook, and analytic code will be made available upon request pending [e.g., application and approval, payment, other].

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
