# Peer review of "Longitudinal Associations Between Maternal Anemia and Breastfeeding Practices on Infant Hemoglobin Levels in the Lao People’s Democratic Republic"

_nutrients, 2025, doi:10.3390/nu17101703_

Round 1

Reviewer 1 Report

Comments and Suggestions for Authors

Dear Author,

The article provide valuable insights into the association between maternal anemia and infant hemoglobin levels, particularly in the context of breastfeeding practices in Lao PDR. 

Below are some suggestions to enhance the article's clarity, structure, and overall impact:

While you mention the burden of anemia, it could be helpful to briefly explain the health implications of anemia in both mothers and infants, such as increased morbidity or developmental delays. This will emphasize the importance of study.

Provide more detailed information about the RCT design, including how participants were selected, the intervention specifics, the control group, and the data collection process.

Discuss the clinical significance of the findings, even if p-values are above conventional thresholds (e.g., p=0.15 and p=0.13). Highlight what these results might imply in real-world settings.

Compare your findings with previous studies, particularly contrasting any differing results. This will help contextualize your findings within the broader research landscape.

Reviewer 2 Report

Comments and Suggestions for Authors

A very interesting analysis, but in my opinion a very significant limitation is the lack of data on serum ferritin levels in both mothers before delivery and in infants during postnatal follow-up. Additionally, the lack of data on the dietary habits of mothers additionally affects the possibility of interpreting the obtained results.

in my opinion the authors should explain why they did not analyze the ferritin level.

The relatively small size of the studied population and the lack of data on ferritin levels are, in my opinion, a major limitation in the possibility of interpreting the obtained results. The assessed manuscript should be supplemented with these data.

Reviewer 3 Report

Comments and Suggestions for Authors

Maternal anemia levels and impact on infant Hb levels are both highly relevant clinical considerations. Consequently, this is a useful and important clinical study.

The Introduction is clear and includes a clear definition of Hb with a clear emphasis on the fact that this issue is a public health challenge. The readers are made aware of who the problems impact and the symptoms. The region where the research took place is clearly highlighted. The study aims are clear. I have no further questions or points to raise about the Introduction.

Materials and Methods: The authors provide a clear explanation of data used and the cohort requirements, including socioeconomic factors. I note that data sets appear to have been used in other studies. Can you please confirm parent consent processes and if signed consent was given or not? 

Exclusions from the study are clear. Statistical analysis is stated and is appropriate for the type of data collected. 

Results: The Results are clearly summarised, with a relevant graph and accessible tables. I have no further comments for this section.

Conclusions: The authors state that maternal anemia does have an impact of infant Hb levels. Can you comment if you were aware of any other factors that could alert healthcare practitioners to be more mindful of low infant Hb levels such as poor infant feeding, cessation of breast feeding, higher number of infant illnesses compared with well peers, etc? 

Limitations are well described, and the fact that the literature for this topic is varied should encourage other healthcare teams to investigate this important area further. 

Round 2

Reviewer 2 Report

Comments and Suggestions for Authors

Thank you to the authors for the response.  The manuscript can be published in its current form.